# Making sense of the pandemic: Parent-child conversations in two cultural contexts

**Pirko Tõugu**[1]*, **Tiia Tulviste**[1], **Lisa Schröder**[2]

**1** Institute of Psychology, University of Tartu, Tartu, Estonia, **2** University of Applied Sciences Magdeburg-Stendal, Magdeburg, Germany

* pirko.tougu@ut.ee

## Abstract

The present study focused on parent-child conversations about COVID-19 related changes in children's lives in Estonia and Germany with an aim to understand how children's conceptual understanding of the disease and their emotional security is created and reflected in these interactions. Twenty-nine parent-child dyads from both cultural contexts provided self-recorded conversations. The conversations were analyzed for the type of explanations, emotional content, and valence. Estonian conversations were longer than those of German dyads. Explanatory talk appeared in both contexts but was general in nature. Conversations in both cultural contexts also included very few emotional references and tended to focus on both positive and negative aspects of the situation. The conversations show that parents tend to support children's coping with stressful situations by helping them conceptually understand COVID-19 and paying little attention to children's comprehension of feelings about the situation.

## Introduction

Children accumulate knowledge and create an understanding about ***the*** world in everyday experiences that often include conversations with more knowledgeable persons [1]. Conversations with parents that include active involvement and explanations support knowledge building [2–4]. In addition, parent-child conversations are an important tool for socio-emotional development [see 5] and carry implications for children's mental health [6]. Therefore, it is important to know how parents talk to their children about the COVID-19 pandemic and the related changes, how they help them create an understanding of the events and pave the way for better coping.

### Cognitive and socio-emotional development support in parent-child conversations

Parent-child conversations serve several functions in child development. According to socio-cultural theory [1], such conversations are an important source of socialization and a tool for knowledge building. Parents scaffold children's budding knowledge and understanding in everyday conversations and socialize children towards culturally appropriate expression and management of emotions [e.g., 7, 8].

**Funding:** This work was supported by the Estonian Research Council grants (PSG296, PI Pirko Tõugu

and PRG1761, PI Tiia Tulviste). The funder had no role in study design, data collection and analysis, decision to publish, or preparation of the manuscript.

**Competing interests:** The authors have declared that no competing interests exist.

Young children's everyday conversations with their parents are a mechanism for remembering [9]. At the same time, these conversations are rich in explanations that help children make sense of real-life events [10–12]. Everyday explanations and parental guidance in exploring complicated things lead to improved knowledge on the matter [2]. Children also actively seek explanations from their parents [13]. Preschool children are more satisfied with and show better memory for explanations as compared to non-explanation responses to their questions and prefer explanations of higher quality over low-quality or scant explanations [14, 15].

Studies of children's concepts show that kindergarten children's understanding of natural phenomena (e.g., plant growth, getting a cold etc.) often reflects both naturalistic or scientific understanding and non-scientific (often animistic or antropomorphic) or intuitive theories [e.g., 16–18]. Intuitive theories about getting sick (e.g., one gets sick due to cold weather) may co-exist alongside scientific theories (e.g., illness is caused by contact with germs) even in adults' understanding and could be more prominent in some cultural contexts than in others [18]. Hernandez et al. [18] also showed that these cultural differences are apparent in parent-child conversations.

Emotion-related conversations are especially important for children's socio-emotional development [e.g., 19, 20] and mental health [6]. Talking about negative emotions that children have experienced could nurture coping and understanding of the causes of emotions, as these conversations usually involve sophisticated discussions of emotions [21, 22]. Such conversations could carry most developmental potential during preschool age [23].

In stressful times, family conversations often hold the key for children's coping. A study focusing on parental explanatory style after the 9/11 attacks has indicated that most parents report providing some fact-based explanations of the event to their pre-teen and teenage children, while others talk about emotions or provide assurance [24]. Regarding explanatory styles, Wilson et al. [24] showed that during stressful times the use of self-focused explanations (e.g., talk about parent's own negative affect or distress) is aversely related to the psychological outcomes of children. Difficulties in family communication (e.g., rumination, discouragement to express one's feelings) have been proposed to be related to negative outcomes for children (e.g., stress, trauma symptoms) during the COVID-19 pandemic [25, 26]. At the same time, ample research indicates that interpretative conversations about negative events often involve discussion and explanation of negative emotions [e.g., 27, 28]. Such discussions have positive implications for children's mental health outcomes [6].

## Children's experience with and understanding of COVID-19

The COVID-19 pandemic has changed the every-day routines and activities for most families around the world. Besides the threat of the illness itself, families are faced with difficulties to cope with changes, restrictions, and insecurities about the future. The impact on families and children has been documented and changes in daily routines [29–33] and mental health [34] reported. Socio-cultural factors, including positive family environment, has been reported to help children and teenagers cope with pandemic related changes [35]. To improve coping, UNICEF has issued guidelines about how to talk to children: these highlight engagement in communication and providing explanations in order to relieve distress in difficult circumstances (UNICEF, 2019). Psychologists and researchers have also called attention to the importance of the opportunities for everyday conversations and parental engagement in children's meaning-making. They recommend parents to (a) provide children with adequate and relevant information, (b) listen for (and curtail sometimes harmful) magical thinking, and (c) participate in emotion-focused conversations [36–38].

There are several studies that have tried to capture children's view and experience of the pandemic. Idoiaga et al. [39] focused on children's conceptualization of the Coronavirus

disease. The authors asked 3-12-year-olds what they associate with the Coronavirus disease and the results indicate that kindergarten children mainly see it as an enemy or a "bad bug" that the doctors are fighting [39]. The study also identified several conflicting feelings that children may experience while being forced to stay at home: safety and happiness on the one hand and boredom, fear, anger, and even loneliness, on the other [39]. Tambling et al. [40] have focused on the topics of pandemic related conversations between parents and children as reported by parents. They found that most often parents reported talking to their children about personal and social hygiene; and about 10% of parents mention children's emotions when probed about children's stress management [40].

Several studies have tried to capture the understanding fostered by parents via collecting parental reports of the questions children pose and responses parents provide [41–43]. These studies show that according to parents, younger children ask more about the changes in everyday life, while older children also want to know about the consequences of the disease [41–43]. About one third of the questions are explanation seeking questions in the US context [42], but only one fifth can be classified as such in the Turkish context [43], suggesting that there could be cultural differences in the questions children ask or parents attend to. Interestingly, Turkish parents provide explanations to their children rather than leaving the questions unattended [43], while in about half of the cases parents in the US do not provide explanations [42]. The responses children receive are realistic but mostly very general [41, 42]. If parents provide more specific responses, then parents of younger children will reference germs in their responses, while parents of older children will be more likely to use illness analogies in their explanations [41]. The responses to children's questions about the reasons for restrictions during the pandemic most often center on the safety and mitigation measures [41–43] and sometimes also authority related explanations. Also, parents often report wanting to shield their children from COVID-19 related information [42, 43].

Prior studies focusing on children's experience and understanding of the pandemic have used parents as informants [40–42]. Parental reports are valuable, but they can only provide an approximation of the actual understanding and experience of the child. To our knowledge there is no comparative observational study investigating parent-child conversations about COVID-19, and no documentation of what the focus of such talk is, and how children's understanding is cultivated by their parents in different cultural contexts. In addition, radical changes and disruptions in everyday routines that have taken place due to the pandemic may give rise to stress and a variety of negative emotions (fear, anxiety, etc.) in children. Parent-child discussion of negative events have been shown to include emotional content. It would be informative to see if parent-child conversations about the pandemic in different cultural contexts address emotions and whether the conversations reflect the recommendations given by experts.

## Present study

The present study focuses on parent-child conversations about the pandemic-related changes in children's life in Estonia and Germany. Despite the importance of parent-child conversations, we do not know how parents in different countries talk to their children about stressful situations, in this case the COVID-19 pandemic. Such information would provide an understanding of the universal characteristics regarding such conversations and the adaptive idiosyncrasies related to cultural styles of talking to children. In the present study, we focus on the explanations regarding the pandemic-related changes in parent-child conversations to understand how parents and children create a conceptual understanding of the pandemic. We also look for the emotional content in the talk with the hope to describe and compare how parents support children's coping and socio-emotional development.

The study focuses on two different Western cultural contexts, Estonia and Germany. According to the theoretical perspectives of different developmental pathways based on the relative importance of two basic human needs–autonomy and relatedness–in socialization of children [44], Germany represents a prototypical Western autonomy-oriented context [44]. In autonomy-oriented contexts socialization of children focuses on their psychological experience, while in relatedness-oriented contexts children are coaxed to see themselves as part of a social system [44, 45]. Parenting in autonomy-oriented cultural context sensitizes children to their subjective mental states such as their wishes, preferences, thoughts, emotions, etc. In this context, children are expected to verbalize and express positive emotions. In relatedness-oriented contexts, children are sensitized to social obligations [46]. Studies show that Estonian mothers promote autonomy in socialization of children as much as German mothers. At the same time, they differ from German mothers by placing more value on relatedness [47, 48], thus representing a psychological relatedness context posed by Kağitçibaşi [45] or autonomy-oriented context where relatedness is more highly valued than in many other typical Western contexts.

Studies of parent-child conversations in these contexts highlight the cultural differences consistent with the cultural models: a pragmatic approach by Estonian parents is revealed in shorter conversations compared to prototypical autonomy-oriented European families, like German or Swedish families, during reminiscing [47, 49] and during mealtime [50]. At the same time, Estonian parents value education highly [51] and seem to be factually oriented in conversations as they ask more information-seeking questions than parents from other European countries [47, 50].

Both Estonian and German mothers and their 4-year-old children [49] and Estonian and Swedish mothers and teens [52] talk little about mental states (incl. emotions). Prior studies in Germany show that mothers refer to emotions when talking about a negative event with their children and provide emotion explanations [53]. However, emotion talk is still more infrequent than in an autonomous related context from Costa Rica.

## Hypothesis

We expect parent-child conversations in both cultures to be rich in explanations and provide some emotion talk. In terms of explanations, general explanations are expected to dominate, but we anticipate both scientific and magical or animistic explanations to appear. Due to the pragmatic approach to conversation, we expect Estonian parent-child conversations to focus more on the knowledge about the disease, description of the changes in children's life, and the explanations for the changes than the German conversations. Compared to Estonian dyads, German dyads are expected to talk more about emotions regarding the pandemic and provide more evaluations of the situation. In addition, we pose a research question regarding the possible negative stress-related emotions that could appear in these conversations: what is the valence of the conversations; do parents and children focus more on the positive or negative aspects of the situation.

## Method

### Sample

Twenty-nine parents from Estonia and 29 from Germany provided audio-recorded conversations with their children during the first or second wave of the pandemic in 2020 and 2021. All German conversations were collected during the first wave; 20 of the Estonian conversations were collected during the first and 9 during the second wave of the pandemic. Conversations collected during the first and second wave in Estonia were compared using a Poisson regression with a log link analysis and the results indicate that the wave predicted the length of the

conversations ($\beta$ = .37, $\chi^2$(1) = 37.11 $p$ < .001); the conversations during the first wave were longer ($M$ = 56.85 clauses) than conversations during the second wave ($M$ = 39.22).

The demographic information about the participants is provided in Table 1. There are no statistically significant differences in the gender distribution or age of the child between the samples. The age of parents did not differ in the two contexts. A Chi-square analysis indicated that there was a significant relationship between cultural context and parental education ($\chi^2$ (2; 58) = 8.90, p < .05); German mothers were more likely to have a Master's degree or above, while more Estonian mothers had a Bachelor's degree. The situation with day-care varied throughout the data collection period: in Germany kindergartens were closed during the lock-down, except for children of front-line workers; the period of lock-down differed according to the geographical region. In Estonia, kindergartens were not officially closed, but children were strongly advised to stay at home during the pandemic. Attending kindergarten was related to the length of the conversation: Poisson regression with a log link analysis indicated that conversations with children attending the kindergarten were shorter that conversation with children staying at home ($\beta$ = -.26, $\chi^2$(1) = 37.24, $p$ < .001).

## Procedure

Parents of preschool children were informed of the study as a call for participants was distributed via social media and university webpages. Participants were asked to provide three conversations with their children and fill in a questionnaire on background information, values, and COVID-19 related changes in the family life and well-being. The parents provided informed written consent to participate in the study, filled in the questionnaire online, and uploaded the conversations via an approved upload page. The present study focuses on one of the conversations where parents were instructed to discuss the changes in children's life during the pandemic and the causes for them. The instructions stressed that they could record the conversation any time that was convenient for them, that they should talk with the child as they normally would, and that the conversation could be as long or short as is natural to them. The prompt for the conversation provided to parents was "Please talk to your child about the changes that have taken place in his/her life after the Coronavirus pandemic started. Also, discuss the reasons for the changes." The procedure used was approved by the Research Ethics Committee of the University of Tartu and University of Applied Sciences Magdeburg-Stendal.

## Coding

The conversations were transcribed, and coders used the MAXQDA program (MAXQDA, 1989–2021) to assign codes to the transcribed conversations. Each main clause by the parent and child was coded using code categories presented in Table 2.

**Table 1. Participants' demographic information.**

| | Estonia | | Germany | |
|---|---|---|---|---|
| | *M (SD)*/ No (frequency) | Range | *M (SD)*/ No (frequency) | Range |
| Age of child (months) | 50 (5.96) | 36–59 | 48 (5.99) | 37–58 |
| Gender of child | | | | |
| Boys (%) | 15 (52%) | | 10 (48%) | |
| Children attending kindergarten (%) | 8 (28%) | | 17 (58%) | |
| Parent's age (years) | 37 (4.27) | 27–46 | 36 (3.90) | 30–45 |
| Parent's education | | | | |
| High-school diploma/ vocational training | 4 (14%) | | 6 (21%) | |
| Bachelor's degree | 17 (59%) | | 6 (21%) | |
| Master's degree | 8 (27%) | | 17 (58%) | |

**Table 2. Coding categories for parent-child conversations.**

| Categories and subcategories | Definition | Examples |
|---|---|---|
| **Information Talk** | Talk about the coronavirus disease | *Parent: What does Coronavirus do?* |
| **Changes Talk** | Talk describing pandemic related changes | *Child: Now you have to wear a mask.* |
| **Explanatory Talk** | Clauses that either provided or asked for explanations. | *P: Why can't you go to kindergarten?*<br>*C: Because if one has Corona, they will pass it to others, and other children get Corona.* |
| Explanatory Talk: general | Explanations that only referenced the disease in general; also, questions for explanations were given this code. | *P: Why can't you go to kindergarten?*<br>*C: You can't go to the kindergarten because there is the disease.* |
| Explanatory Talk: scientific | Explanations that refer to the mechanisms of disease transmission | *P: Because germs come out of our nose and mouth when we talk, and masks help us not to spread germs.*<br>*C: Because you could get infected.* |
| Explanatory Talk: animistic | Explanations that prescribe human characteristics and aspirations to the virus | *C: Corona looks for our mouth to come in.* |
| Explanatory Talk: other | These included pragmatic explanations, socially oriented explanations, and explanations focusing on rules. | *C: (I am not going to kindergarten), because you are at home.*<br>*C: We don't go out, because otherwise we meet friends and give them hugs.*<br>*C: (I am not going to kindergarten), because it is not allowed.* |
| **Evaluative talk** | Clauses that provided personal evaluations of the situation | |
| Positive | Clauses providing positive evaluations. | *C: I think it's nice to be inside home and play with Legos.* |
| Negative | Clauses providing negative evaluations. | *C: It is boring.* |
| Undefined | These included evaluations or questions for evaluations that could not be classified as positive or negative. | *P: What do you think of being at home?* |
| **Emotion Talk** | Talk that referred to emotions regarding the situation. | |
| Positive | Talk that referred to positive emotions regarding the situation. | *P: And this makes you happy?* |
| Negative | Talk that referred to negative emotions regarding the situation. | *P: Are you afraid of Corona?* |
| Undefined | This category included questions about emotions and feelings that could not be categorized as positive or negative. | *P: How do you feel in this situation?* |
| **Other Talk** | This category included utterances about life in general or the situation at hand. | *P: Do you like to go the playground?*<br>*C: Let's play!* |

The coding system was exhaustive and exclusive; two coders coded 20% of the Estonian and German transcripts, intercoder reliability was $\kappa = .80$ for child utterances and $\kappa = .84$ for parent utterances.

## Results

For the hypothesis testing count numbers of the different code categories were used and Poisson regression models with a log function were constructed to identify the effect of context. Maternal education was included as a covariate to control for the differences in the conversation that could be ascribed to maternal education. For the exploratory analyses, the ratio of negative and positive evaluations to all evaluations and the ratio of negative and positive emotions to all emotions by the dyad was calculated. Nonparametric tests were used to compare the ratio measures of the Estonian and German context.

### Descriptive analyses

The average length of the coded conversations was 44 ($SD = 28.54$, range 6–165) main clauses, with parents uttering 29.27 ($SD = 19.58$) clauses on average and children contributing 14.66 ($SD = 10.59$) clauses. The largest part of the conversations was devoted to pandemic related changes in both contexts ($M_{clauses} = 28.12$, range 2–69). The means and standard deviations for all the coded categories in the two contexts are presented in Table 3.

**Table 3. The means, standard deviations, and proportion of talk for all the coded categories by cultural context.**

| Category | Estonian dyads | | German dyads | |
|---|---|---|---|---|
| | Mean number of clauses *(SD)* | Proportion of the talk devoted to the category | Mean number of clauses *(SD)* | Proportion of the talk devoted to the category |
| Information talk | 6.34 (7.43)*** | .12 | 3.21 (4.62)*** | .08 |
| Changes talk | 32.10 (18.52)*** | .65 | 24.14 (14.91)*** | .63 |
| Explanatory talk (total) | 5.21 (6.24) | .12 | 4.55 (5.60) | .12 |
| *General explanations* | 3.38 (4.79) | | 3.17 (4.01) | |
| *Scientific explanations* | .48 (.91) | | .86 (1.33) | |
| *Animistic explanations* | .07 (.25) | | .00 (.00) | |
| *Other explanations* | 1.28 (1.96) | | .52 (1.24) | |
| Evaluative talk (total) | 3.79 (5.99)*** | .07 | 6.86 (9.09)*** | .16 |
| *Negative* | .93 (2.22) | | 2.62 (3.85) | |
| *Positive* | 1.97 (2.69) | | 1.79 (3.24) | |
| Emotion talk (total) | 3.93 (9.20) | .04 | .69 (1.79) | .02 |
| *Negative* | 1.69 (3.81) | | .51 (1.77) | |
| *Positive* | .59 (1.86) | | .14 (.44) | |
| Total no of main clauses | 51.38 (34.25)*** | 1.0 | 39.45 (22.36)*** | 1.0 |

***- statistically significant effect of context ($p < .001$) detected using Poisson regression with a log link.

Note: The Estonian sample includes an outlier; the results remained the same when analyses were run with and without the participant.

First, separate Poisson regressions with a log function were used to establish the effect of context (Estonia vs Germany) on the total number of main clauses, the number of clauses devoted to Information talk, Changes talk, Explanatory talk, and Evaluative talk. Context predicted the total number of main clauses ($\beta = 3.68$, $\chi^2(1) = 45.58$ $p < .001$), the number of clauses devoted to Information talk ($\beta = .77$, $\chi^2(1) = 35.06$, $p < .001$), and Changes talk ($\beta = .29$, $\chi^2(1) = 31.54$, $p < .001$). Estonian dyads allotted more main clauses to these topics than German dyads. Evaluative talk was also predicted by the context ($\beta = -.59$, $\chi^2(1) = 24.90$, $p < .001$) with German dyads devoting more utterances to evaluative talk than Estonian dyads. The amount of Explanatory talk did not differ between the contexts. Maternal education was a significant covariate for the total number of main clauses ($\beta = .10$, $\chi^2(1) = 11.83$, $p < .001$), Information talk ($\beta = .32$, $\chi^2(1) = 12.13$, $p < .001$), and Evaluative talk ($\beta = .19$, $\chi^2(1) = 5.66$, $p < .05$).

As emotion talk was very infrequent with 72% of Estonian and 76% of German dyads not mentioning emotions, a categorical variable was built for this measure (dyads mentioning emotions vs dyads not mentioning emotions). A Chi-square analysis was run to see if the context was related to the dyad's use of emotions. The use of Emotion talk did not differ between the contexts.

For the exploratory analyses, ratios of positive and negative evaluations to total number of evaluations were calculated to investigate the valence of the talk. Separate Mann-Whitney U tests were run to investigate the differences between contexts in the negative and positive evaluation ratios. The ratio of negative evaluations did not differ between the contexts, but the ratio of positive evaluations did ($U = 93.5$, $p = .007$): Estonian dyads used more positive evaluations relative to all evaluations than German dyads.

## Discussion

Parent-child conversations about the COVID-19 pandemic related changes in children's life and the reasons for the changes were studied for their focus and valence in Estonia and Germany. Specifically, we were interested in the types of explanations regarding COVID-19 related changes and the naturally occurring emotion talk concerning the pandemic.

As expected, the conversations did contain explanations, including scientific ones, but contrary to our expectations, only a couple of animistic explanations appeared. Studies show that regarding falling ill, people often refer to intuitive theories (e.g., one gets a cold due to being cold) [18]. There are cultural differences in children's biological explanations [54] and in cause of illness understanding [18] indicating that some cultural contexts are more prone to some intuitive theories. Earlier studies also indicate that young children make sense of natural phenomena using magical or animistic thinking [55, 56]. Prior studies of explanations regarding the Coronavirus show that parents report providing correct realistic responses to children's questions in different cultural contexts [41–43]. Haber et al. [41] also showed that parents of 3-5year-olds often reference germs.

In the present study, both parents and children provide correct, albeit general information regarding the disease and the pandemic-related changes. The present study included mostly highly educated parents and their children of two European contexts and maternal education was not a significant covariate for the use of explanations. It appears that all parents were rather well-informed of the causes of the pandemic and the transfer mechanisms of the disease; the conversations with their children seem to reflect that as they do not resort to animistic explanations or intuitive theories. It is also possible that the findings reflect the tendency of parents with strong autonomous orientation to support children's cognitive development [44, 45]. In any case, it appears that regarding pandemic-related explanations, parents in these two Western contexts have managed to curtail the sometimes-harmful magical thinking that experts warn against [e.g., 36, 37].

Similar to the results of Haber et al. [41], most explanations in the conversations were general in nature. One possible reason is that parents consider scientific explanations to be beyond their children's grasp. And prior research has shown that it is very difficult for adults to appropriate scientific explanations to children's level of comprehension [57]. Also, general explanations could dominate among explanations because it was often the parent who asked for the explanation and the child that provided one. Few children in our sample were asking for the reasons behind specific changes. After children provided an explanation, mothers often did not elaborate further. May be to let the child decide on how much they want to focus on the matter or considering general explanations to be appropriate for children of that age. The following excerpts are typical examples:

C: Eh, I know what we are not allowed to do. Namely, we are not allowed to cuddle with each other.

P: Mhm. With nobody?

C: But only with our families.

P: Mhm. And do you know why?

C: Because otherwise, because we don't know whether they have Corona.

P: Hm, do you know why you stayed home for so long?

C: Because of the virus.

P: Because of the virus, exactly. Do you remember what's going on with the virus?

C: All people sick.

P: Exactly. A lot of people got sick. And because of that?

C: Because of that?

P: You weren't allowed to go to the kindergarten, right?
C: Right.
P: And what else happened? Who wasn't allowed to go to work in the beginning?
C: Mom.
(German parents and children)

Interestingly, a study by Menendez et al. [42] conducted in the US showed that parents of children aged 3–12 report providing mostly socially oriented explanations for the changes related to the pandemic. This contradicts the present study where we see that the majority or explanations discussed are either general (e.g., "Because of the disease.") or scientific that provide information about how the disease spreads. Here, the reason for such difference could be the method of research: Menendez et al. [42] use parental reports where parents provided a selected sample of questions. The difference could also appear due to the different age groups studied: the sample of Menendez et al. [42] also included older children whose social circle is wider and who could therefore request more information and explanations regarding social restrictions. Finally, it would be interesting to see if observational studies with kindergarteners in the US would have the same focus.

In general, the Estonian and German conversations seemed rather similar, indicating that parents approached the task in a similar manner. At the same time, differences in volume appeared: Estonian conversations were longer as parents and children talked more about the Coronavirus itself and the related changes than German parents and children. While in other situations such as talking about personal experiences parent-child talk in Estonia has been shown to be brief compared to other European contexts [49, 50], the result is surprising. It could, perhaps, be ascribed to the pragmatic perspective on conversations in Estonia attributed to the autonomy-relatedness oriented context: it is possible that Estonian parents feel comfortable talking to children about actual rather than personal matters. Estonians have been found not to be very good in small talk, they speak when there is something important to say, and tolerate silence more than people from many other nations [58]. It also seemed that they would not try to shield the child from unpleasant topics; the conversations featured serious consequences, being ill, and even death, as illustrated in the following excerpt:

P: But why do we have to stay home?
C: Because there is Corona.
P: But why is it bad? It is a bad disease.
C: Yes, and then people die.
P: People die, yes. Did you have Corona too or?
C: Yes.
P: Who else had it?
C: You and daddy.

These findings resonate with prior studies conducted in different cultural contexts. Studies focusing on prototypical autonomy-oriented contexts have shown that parents would want to shield children from pandemic related information [42] or not focus on the unfortunate consequences of the pandemic with young children [41]. A study conducted in Turkey [43] that is a more relatedness-oriented context showed that parents and children focused more on the virus itself rather than life-style changes.

German parent-child dyads were expected to discuss the emotional effect of the pandemic more that Estonian dyads, but this was not the case. Instead, all participants talked very little about emotions. This is surprising, as from the theoretical viewpoint, it is developmentally a good time to discuss emotions with children in order to support their socio-emotional development [e.g., 20, 21]. From the practical viewpoint, discussing emotions at difficult times is important for better coping and mental health [6, 38]. The present study did not prompt for

emotion talk and prior conversational studies focusing on other situations suggest that emotion talk is not very prominent [49, 53]. Also, Tambling et al. [40] showed that when parents were surveyed about managing children's stress, only about ten percent of parents mentioned emotions. Yet, it is still surprising that even in stressful situations, when lives of families have been turned upside down, the talk about emotions with children is very sparse.

Although studies suggest that preschool age is an important period of children's lives to discuss emotional reactions [23], it is possible that children of this age group are not as much emotionally or at least negatively affected by the pandemic as perhaps older children and parents. A closer look at the conversations reveals that a considerable proportion of emotion and evaluative talk is devoted to positive aspects of the situation. The positive aspect is illustrated in the following excerpt:

P: But did you miss your friends?

C: No. I just think that it is fun to be at home and fun to play at home.

(Estonian mother and child)

The negative aspects mentioned most often revolved around boredom and not fear or anxiety. An excerpt-example on boredom:

P: . . . But the things we are doing every day, don't you like them too, are they bad too?

C: Boring.

P: Boring? You are bored? What is boring? Tell me.

(German parent and child)

It is also important to note that maternal education was a significant covariate for the length of the conversations, for talking about Corona virus in general, and for providing evaluations of the situation: children of more educated mothers participated in conversations that were longer, included more information about the disease, and discussed the pros and cons of the situation more often than children of less educated mothers in both contexts. Maternal education has been identified as a crucial factor influencing input in language development [59], therefore future studies of children's experiences could also pay attention to differences linked to maternal education.

The present study focused on conversations between preschool children and their parents about the pandemic related changes and reasons for the changes. The study is not without limitations, the most significant being that the study was voluntary and included mostly highly educated parents. It is probable that parents or children who are really struggling with coping in the new reality opted out of the study. Salmon [38] has pointed out that the pandemic could be especially detrimental to more vulnerable families and further research should make an effort to include a wider range of parents and children. Future studies could also clarify if the findings generalize to more educationally diverse samples. In addition, the study had a rather small sample and focused on one conversation that parents provided: future studies should also try to capture naturally occurring talk and expand the sample. As parental conversations are related to children's language development [59], it would be advisable to include a measure of child language development.

Nevertheless, the results show that parents support children's conceptual understanding of the pandemic and hence cognitive development as they discuss the disease and the reasons for changes with their children. The conversations showcase factually correct (not animistic or magical), yet rather general notions of the disease and the pandemic related changes. Surprisingly, little attention is paid to the emotions and feelings about the stressful situation and hence socio-emotional development. The results carry practical implications: educated parents in European contexts have a good grasp on supporting cognitive understanding of current situations. At the same time, they could use a reminder to also focus on discussing emotions regarding the situation in order to support socio-emotional development and children's coping in stressful situations.

## Supporting information

**S1 File. An anonymized dataset of parent-child conversations about COVID-19 in Estonia and Germany.**
(SAV)

## Author Contributions

**Conceptualization:** Pirko Tõugu, Tiia Tulviste, Lisa Schröder.

**Formal analysis:** Pirko Tõugu.

**Funding acquisition:** Pirko Tõugu.

**Methodology:** Pirko Tõugu, Tiia Tulviste, Lisa Schröder.

**Project administration:** Lisa Schröder.

**Software:** Pirko Tõugu, Lisa Schröder.

**Writing – original draft:** Pirko Tõugu.

**Writing – review & editing:** Tiia Tulviste, Lisa Schröder.

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
