## [Decision Letter · Decision Letter 0]

7 Sep 2022

PONE-D-22-19799Making Sense of the Pandemic: Parent-Child Conversations in Two Cultural ContextsPLOS ONE

Dear Dr. Tõugu,

Thank you for submitting your manuscript to PLOS ONE. After careful consideration, we feel that it has merit but does not fully meet PLOS ONE’s publication criteria as it currently stands. Therefore, we invite you to submit a revised version of the manuscript that addresses the points raised during the review process.

 Can you please address the concerns raised by the two expert reviewers?

We look forward to receiving your revised manuscript.

Kind regards,

Avanti Dey, PhD

Staff Editor

PLOS ONE

Journal Requirements:

"This work was supported by the Estonian Research Council grant (PSG296)."

"This work was supported by the Estonian Research Council grant (PSG296), (PI Pirko Tõugu). The funder had no role in study design, data collection and analysis, decision to publish, or preparation of the manuscript."

Reviewers' comments:

Reviewer's Responses to Questions

**Comments to the Author**

1. Is the manuscript technically sound, and do the data support the conclusions?

Reviewer #1: Partly

Reviewer #2: Partly

2. Has the statistical analysis been performed appropriately and rigorously? 

Reviewer #1: I Don't Know

Reviewer #2: Yes

3. Have the authors made all data underlying the findings in their manuscript fully available?

Reviewer #1: No

Reviewer #2: No

4. Is the manuscript presented in an intelligible fashion and written in standard English?

Reviewer #1: Yes

Reviewer #2: Yes

5. Review Comments to the Author

Reviewer #1: PONE-D-22-19799

I think this study is interesting and has merit in terms of shedding light on parent-child conversations and how these may vary across contexts that seem similar at first (two Western nations). At the same time I have some serious concerns about the framing of the paper. The study analyzed a single parent-child conversation surrounding the Coronavirus -- “the changes in children’s life during the pandemic and the causes for them”. The authors frame the study as examining a naturalistic conversation. I am not certain that this is really the case given the nature of the prompt for the parents (there is no way to really assess how regularly these types of conversations occur) nor the reliance on parents having to record the conversations, which is certainly subject to social desirability bias.

The literature review needs to be elaborated, both in terms of research on parent-child talk and interactions during Covid-19 (see some of the studies below), as well as the section on talk during stressful times. The authors need to strengthen their case for why to look particularly at conversations during this time as a way to examine the particular characteristics of these conversations. For example, are these conversations more or less likely to include emotion-related talk?

Please see some of the following published studies, which should be included in the research relating to parent-child talk surrounding Covid-19:

Tambling, R. R., Tomkunas, A. J., Russell, B. S., Horton, A. L., & Hutchison, M. (2021). Thematic analysis of parent–child conversations about covid-19:“playing it safe”. Journal of child and family studies, 30(2), 325-337.

Haber, A. S., Kumar, S. C., Puttre, H., Dashoush, N., & Corriveau, K. H. (2022). “Why Can't I See My Friends and Family?”: Children's Questions and Parental Explanations About Coronavirus. Mind, Brain, and Education, 16(1), 54-61.

Israel, O. M. E. P., Dorit, A., Tom, C. C., Galia, M. K., & Laly, M. Shared Book Reading at Home and at School Prior to and Since the COVID-19 Outbreak.

Aram, D., Asaf, M., Karabanov, G. M., Ziv, M., Sonnenschein, S., Stites, M., ... & López-Escribano, C. (2022). Beneficial Parenting According to the “Parenting Pentagon Model”: A Cross-Cultural Study During a Pandemic. In The Impact of COVID-19 on Early Childhood Education and Care (pp. 215-236). Springer, Cham.

Methods:

• More information about the children is needed – are they in some kind of preschool setting (and does this vary between Estonia & Germany)? What was the state of these settings during the time of data collection (lockdown, etc.).

• The authors note that data was collected during the first and second wave of the pandemic. How many conversations were collected during each wave? Was any assessment done to examine differences in conversations between the time points?

• Were the children evaluated for language development? This can significantly impact how parents talk with their children.

• What was the prompt given to parents in terms of the conversation?

Results:

• GLM for Poisson distribution were run – were these negative binomial regressions? If so, please specify. If not, please specify exactly which type of regressions were run.

• The authors assessed length by number of total clauses, but as parent-child talk is very variable by individual, was this made into some kind of ratio or presented in some other way that might account for this variability? All that this will tell us otherwise is that some parents speak more than others (plus, as mentioned above, there was no accounting for children’s language development).

Discussion:

• The authors note cultural differences relating to types of explanations (scientific, animistic, etc.). Have any of these been examined specifically in the Estonian/German contexts? If so, how do your results here support/contradict those? If no studies have been done in these contexts, then I think some additional exploration or thoughts of why these types of explanations might be common in these contexts is warranted here.

• I believe that the limitations of the study need to be expanded. This was a study conducted with a fairly small number of highly educated parents, examining a single conversation. This should be acknowledged as an important limitation. The need to further examine conversation of various types is necessary. Another significant limitation is the lack of any kind of information about the children’s language ability, which can certainly impact the nature of parent-child conversations. This must be acknowledged.

Editing:

• Be consistent in terminology – sometimes you write Coronavirus (one word) and sometimes Corona virus (two words).

• The article should be carefully edited for English language – commas, subject-verb agreement, etc.

Reviewer #2: This is a well written manuscript investigating an important topic namely how parents actually talk with their children about the COVID-19 pandemic. An additional strength is the cultural comparison. The Introduction sets up the study very well, and I particularly liked the writing style, which was engaging and remarkably jargon-free.

Comments:

1. It would have been useful to have more information on the differences and similarities between the two comparison cultures (Estonia, Germany) to set up hypotheses regarding cultural differences in reminiscing. The authors suggest that these cultures differ “slightly” in their focus on autonomy vs relatedness, and parent socialisation factors reflect these differences. How different is “slight” and how might this be manifested more generally in each culture?

2. Relatedly, as the authors acknowledge, both samples were quite small, and both were highly educated. How were the samples recruited? Were the samples sufficiently large and diverse to detect “true” cultural differences?

3. More information about the approach to coding would be appreciated. How does the MAXDA programme operate? What is the role of human coders vs the programme?

4. How were the particular conversational examples selected for the Discussion.

5. There are some presentation issues: the list format of the coding in the Method; the use of the words “of which” in Table 1 (it is unclear what this means).

The following manuscript that might be helpful in placing parents’ response to the pandemic in a broader context, and includes a focus on parent-child reminiscing, might be useful.

Salmon, K. (2021). The ecology of youth psychological wellbeing in the COVID-19 pandemic. Journal of Applied Research on Memory and Cognition,10,(4), 564-576 .

6. PLOS authors have the option to publish the peer review history of their article (what does this mean?). If published, this will include your full peer review and any attached files.

Reviewer #1: No

Reviewer #2: No

---

## [Author Response · Author response to Decision Letter 0]

12 Oct 2022

Response letter to the reviewers

Thank you very much for reviewing the manuscript and pointing out important caveats in the form it was first submitted. We have tried to address all of your concerns and we hope that you find the manuscript much improved. I will respond and point out the amendments point by point below.

Concerns raised by the academic editor:

1. The style of the manuscript (titles, references etc.) was aligned with the requirements of PLOS ONE.

2. The Acknowledgement section was deleted. As it was, it only doubled the information already presented in the Funding Section. The Funding Section contains all necessary information and does not need to be updated.

3. The anonymized data set is uploaded.

4. The method section is updated with information about written consent and the ethics committees that oversighted the project.

Reviewer 1

The introduction and especially the part about Corona experience was updated and carefully reworked in order to point out the importance of the manuscript (observational comparative study) yet be respectful of its limitations (not naturally occurring talk, but self-selected recordings). We have also taken care to clarify why emotions are expected to be mentioned.

In the method section:

- More information is provided about the preschool setting during data collection.

- It is clarified how much data was collected during the first and the second wave.

- Unfortunately, children’s language abilities were not assessed in this study. We do now refer to the need to do so in future studies in the limitation section.

- The prompt provided to parents is now presented in the method section.

In the results section:

- Poisson regression with the log link were run to analyze the data, it is now also clarified in the manuscript.

- The mean number of clauses for children and parents are also now provided in the Descriptive analyses section. In addition, proportions of the type of talk relative to the amount of talk are presented in Table 2.

Discussion:

- Unfortunately, the types of explanations for illness have not been studied in these contexts before. At the same time, we have rewritten the introduction and discussion section to provide a better foundation for hypothesis and the explanation for the results.

- The limitation of the study were expanded; both the restrictions impinged by the sample and the fact that we do not have language assessment data are highlighted.

In general:

- Inconsistences in terminology were removed.

- The manuscript was carefully proofread.

Reviewer 2

1. We rewritten the instruction (and especially “Present study” section) where we now provide more information about the slightly different focus on autonomy and relatedness and the related differences in socialization.

2. More information is provided about recruitment and the small sample size and need for more diversity is acknowledged in the limitation section of the manuscript

3. The MAXQDA program was used to manually assign codes to segments of the transcript. No automatic coding was used. This is now also clarified in the manuscript.

4. Examples were selected from among other similar ones to display the particular coding category under discussion and to illustrate the results. 

5. Presentation issues were fixed, and the suggested literature added to the introduction.

We thank you again for your input and hope you find the manuscript improved!

---

## [Decision Letter · Decision Letter 1]

14 Nov 2022

PONE-D-22-19799R1Making Sense of the Pandemic: Parent-Child Conversations in Two Cultural ContextsPLOS ONE

Dear Dr. Tõugu,

Thank you for submitting your manuscript to PLOS ONE. After careful consideration, we feel that it has merit but does not fully meet PLOS ONE’s publication criteria as it currently stands. Therefore, we invite you to submit a revised version of the manuscript that addresses the points raised during the review process.

The manuscript has been evaluated by two reviewers, and their comments are available below.

The reviewers have raised a number of concerns and clarifications required regarding the sample, the examples of coding categories, some of the statistical results, and the discussion. 

We look forward to receiving your revised manuscript.

Kind regards,

Alice Coles-Aldridge

Editorial Office

PLOS ONE

Reviewers' comments:

Reviewer's Responses to Questions

**Comments to the Author**

1. If the authors have adequately addressed your comments raised in a previous round of review and you feel that this manuscript is now acceptable for publication, you may indicate that here to bypass the “Comments to the Author” section, enter your conflict of interest statement in the “Confidential to Editor” section, and submit your "Accept" recommendation.

Reviewer #1: (No Response)

Reviewer #2: (No Response)

2. Is the manuscript technically sound, and do the data support the conclusions?

Reviewer #1: Yes

Reviewer #2: Partly

3. Has the statistical analysis been performed appropriately and rigorously? 

Reviewer #1: Yes

Reviewer #2: Yes

4. Have the authors made all data underlying the findings in their manuscript fully available?

Reviewer #1: Yes

Reviewer #2: Yes

5. Is the manuscript presented in an intelligible fashion and written in standard English?

Reviewer #1: Yes

Reviewer #2: Yes

6. Review Comments to the Author

Reviewer #1: I appreciate the changes the authors have made, particularly the expanded explanations of the contexts between the two countries. I still have some questions/clarifications before I think the article can be published:

- Regarding the sample:

o Did you run any kind of t-tests regarding education? There seem to be many more German mothers with master’s degrees. Was this a significant difference? Since we know education can impact conversations, this has the potential to impact the results.

o Similarly, was any evaluation done on the conversations between those who were in kindergarten compared to those who were not? The ones who were might also have been getting information from kindergarten staff or hearing other explanations, which might have made a difference.

- It would be helpful if the examples in the table of the coding categories provided examples of child utterances as well as parents, or at least clearly distinguished which is which.

- The second table (descriptive statistics of the conversations) should be labeled Table 2.

o At the bottom of the table, you note that the * represents statistical significance at p<.001, but traditionally, a single star denotes significance at p<.05. If indeed significance was at p<.001, you should use three asterisks to denote this (or whatever alternative the journal format calls for).

- For the Mann-Whitney results, you also noted p<.005, but this is an unusual reporting. Did you mean .05 or .001? Or was p=.005? Please clarify.

- In the Discussion, you refer to Menendez et al., but do not note the context, which may be relevant. Please add this.

- There is still some editing that needs to be done (e.g., the first sentence of the abstract has grammatical mistakes).

Reviewer #2: This manuscript is much improved and I congratulate the authors on their comprehensive approach to the revisions.

I have a couple of outstanding comments.

1. The number of boys and girls differs between cultural samples. As reminiscing style is, at least at times, influenced by gender, it would be good to see greater consideration of this factor in the Discussion and Results. For example, do the results remain the same when child gender is a control variable?

2. There could be greater nuance in the discussion of cultural differences. For example, in the Introduction (p.6) there is a statement made about the difference between US and Turkish samples - yet the US (and possibly Turkey) is comprised of myriad subcultures and I"m wondering if it's possible to generalise in this way. Perhaps the authors could describe their samples more - they talk about the high education in both their German and Estonian samples in the Discussion, which is helpful.

7. PLOS authors have the option to publish the peer review history of their article (what does this mean?). If published, this will include your full peer review and any attached files.

Reviewer #1: No

Reviewer #2: No

---

## [Author Response · Author response to Decision Letter 1]

29 Nov 2022

Thank you very much for your careful attention to the manuscript and the points that you raised! We have addressed all your concerns and hope you find the manuscript improved! Here are the changes we made in respect to reviewers’ comments:

Reviewer 1:

• As suggested by reviewer 1 Analyses were run regarding the parental education of the sample. It turned out the German mothers are more likely to have a Master’s degree or above, while Estonian mothers are more likely to have a Bachelor’s degree. A note about possible effects of this educational difference is also added to the limitations section. 

• Also, analyses were run in order to establish if attending kindergarten was related to the length of the conversations and the results are reported in the manuscript. 

• It is now indicated whether the utterance is provided by the child or parent in the Table. It is not possible to provide examples by parent and child at all times, as not all categories were used by both e.g. children did not use Evaluative Undefined category or parents did not use Animistic Explanations.

• The error in the heading of the second table was fixed and it is Table 2 in the revised version of the manuscript. The reference in the text is also fixed.

• Asterix were added to the values to adhere to reporting conventions.

• The mistake is fixed and an exact p value for Mann-Whitney is reported in the manuscript.

• The context where the Menendez et al study was carried out is included in the Discussion.

• The manuscript was carefully proof-read.

Reviewer 2:

• The question of gender difference in parent child conversation is an interesting question in itself. But in the current study, the difference in the number of girls and boys in the two samples is not statistically significant. Therefore, there is no reason to include gender as a control variable, especially considering the small sample size.

• We have included the information about parent age in the manuscript in order to describe the sample more. The sample demographics are now also organized in a table to provide a better overview. We originally also asked about household size, but something got lost in the translation, and we unfortunately do not have that data to provide.

Thank you very much for your valuable input and for helping us improve the quality of the manuscript!

---

## [Decision Letter · Decision Letter 2]

19 Dec 2022

PONE-D-22-19799R2Making Sense of the Pandemic: Parent-Child Conversations in Two Cultural ContextsPLOS ONE

Dear Dr. Tõugu,

Thank you for submitting your manuscript to PLOS ONE. After careful consideration, we feel that it has merit but does not fully meet PLOS ONE’s publication criteria as it currently stands. Therefore, we invite you to submit a revised version of the manuscript that addresses the points raised during the review process. Your paper has been re-reviewed. Although the comments received from our referees are overall positive, one of them asks for a couple of minor revisions. Could you please address them to the best of your ability and resubmit? If the amendments are outstanding, I will proceed to make an editorial decision without the need for a further round of reviews.

We look forward to receiving your revised manuscript.

Kind regards,

Sergio A. Useche, Ph.D.

Academic Editor

PLOS ONE

Journal Requirements:

Reviewers' comments:

Reviewer's Responses to Questions

**Comments to the Author**

1. If the authors have adequately addressed your comments raised in a previous round of review and you feel that this manuscript is now acceptable for publication, you may indicate that here to bypass the “Comments to the Author” section, enter your conflict of interest statement in the “Confidential to Editor” section, and submit your "Accept" recommendation.

Reviewer #1: (No Response)

Reviewer #2: All comments have been addressed

2. Is the manuscript technically sound, and do the data support the conclusions?

Reviewer #1: Partly

Reviewer #2: Partly

3. Has the statistical analysis been performed appropriately and rigorously? 

Reviewer #1: I Don't Know

Reviewer #2: Yes

4. Have the authors made all data underlying the findings in their manuscript fully available?

Reviewer #1: Yes

Reviewer #2: Yes

5. Is the manuscript presented in an intelligible fashion and written in standard English?

Reviewer #1: Yes

Reviewer #2: Yes

6. Review Comments to the Author

Reviewer #1: I appreciate the authors addressing the previous round of comments. Regarding the significant differences in parents' education between contexts - I don't see how you can add this to the limitations but not control for it in the regression analyses. It would make sense that the type and content of conversation would relate to education. If there were significant differences between the context on this variable, then comparing the contexts without taking this variable into consideration is problematic.

Reviewer #2: (No Response)

7. PLOS authors have the option to publish the peer review history of their article (what does this mean?). If published, this will include your full peer review and any attached files.

Reviewer #1: No

Reviewer #2: No

---

## [Author Response · Author response to Decision Letter 2]

28 Dec 2022

We would also like to thank the Reviewer for their suggestion. We have changed the manuscript as follows:

- We have taken into account the suggestion by Reviewer 1 and run the regression analyses with maternal education as a covariate. The results are provided in the manuscript. We also added a paragraph to the Discussion to refer to the results of these analyses. 

Thank you for having our work reviewed!

---

## [Editor Report · Decision Letter 3]

8 Jan 2023

Making Sense of the Pandemic: Parent-Child Conversations in Two Cultural Contexts

PONE-D-22-19799R3

Dear Dr. Tõugu,

We’re pleased to inform you that your manuscript has been judged scientifically suitable for publication and will be formally accepted for publication once it meets all outstanding technical requirements.

Kind regards,

Sergio A. Useche, Ph.D.

Academic Editor

PLOS ONE

Additional Editor Comments (optional):

Thanks for the amendments made. The paper can now be accepted in its present form.

---

## [Editor Report · Acceptance letter]

13 Jan 2023

PONE-D-22-19799R3 

Making Sense of the Pandemic: Parent-Child Conversations in Two Cultural Contexts 

Dear Dr. Tõugu:

I'm pleased to inform you that your manuscript has been deemed suitable for publication in PLOS ONE. Congratulations! Your manuscript is now with our production department. 

Kind regards, 

on behalf of

Dr. Sergio A. Useche 

Academic Editor

PLOS ONE